# Detailed Characterization of the Lung–Gut Microbiome Axis Reveals the Link between PD-L1 and the Microbiome in Non-Small-Cell Lung Cancer Patients

**DOI:** 10.3390/ijms25042323

**Published:** 2024-02-15

**Authors:** Vytautas Ankudavicius, Darja Nikitina, Rokas Lukosevicius, Deimante Tilinde, Violeta Salteniene, Lina Poskiene, Skaidrius Miliauskas, Jurgita Skieceviciene, Marius Zemaitis, Juozas Kupcinskas

**Affiliations:** 1Department of Pulmonology, Lithuanian University of Health Sciences, LT-44307 Kaunas, Lithuania; 2Institute for Digestive Research, Lithuanian University of Health Sciences, LT-44307 Kaunas, Lithuania; 3Department of Pathology, Lithuanian University of Health Sciences, LT-44307 Kaunas, Lithuania; 4Department of Gastroenterology, Lithuanian University of Health Sciences, LT-44307 Kaunas, Lithuania

**Keywords:** lung cancer, lung–gut axis, microbiota, PD-L1

## Abstract

Next-generation sequencing technologies have started a new era of respiratory tract research in recent years. Alterations in the respiratory microbiome between healthy and malignant conditions have been revealed. However, the composition of the microbiome varies among studies, even in similar medical conditions. Also, there is a lack of complete knowledge about lung–gut microbiome interactions in lung cancer patients. The aim of this study was to explore the lung–gut axis in non-small-cell lung cancer (NSCLC) patients and the associations between lung–gut axis microbiota and clinical parameters (CRP, NLR, LPS, CD8, and PD-L1). Lung tissue and fecal samples were used for bacterial 16S rRNA sequencing. The results revealed, for the first time, that the bacterial richness in lung tumor tissue gradually decreased with an increase in the level of PD-L1 expression (*p* < 0.05). An analysis of β-diversity indicated a significant positive correlation between the genera Romboutsia and Alistipes in both the lung tumor biopsies and stool samples from NSCLC patients (*p* < 0.05). Survival analysis showed that NSCLC patients with higher bacterial richness in their stool samples had prolonged overall survival (HR: 2.06, 95% CI: 1.025–4.17, *p* = 0.0426).

## 1. Introduction

The human microbiome is composed of thousands of bacterial species that colonize the gut, skin, oral cavity, urinary tract, and reproductive system [1,2]. Our microbiome is shaped by multiple factors, including diet, the environment, antibiotics, and other factors [3]. Multiple studies have suggested that deregulation of the human microbiome may contribute to the development of various complex diseases, including cancer [4,5,6,7,8,9].

For a long time, the lungs were considered sterile organs until culture-independent technologies revealed that they are inhabited by diverse microbial communities [10]. Early reports based on culture-dependent methods linked *Chlamydia pneumoniae* and *Mycobacterium tuberculosis* with the risk of lung cancer [11,12]. Advanced analyses of the lower respiratory tract microbiome began in the past decade, and have demonstrated distinct microbiome alterations between healthy and malignant respiratory tract conditions [13,14,15]. Several studies reported that *Veillonella*, *Megasphera*, *Streptococcus*, *Enterobacter*, and *Legionella* may be associated with lung cancer development [16,17,18]. However, the mentioned species have also been reported as part of a normal lung microbiome in healthy individuals [19,20,21].

Lung cancer treatment can vary depending on the cancer type and stage, and the patient’s overall health condition. The most common methods for lung cancer treatment, including for non-small-cell lung cancer (NSCLC), include surgery, radiotherapy, chemotherapy, targeted therapy, and immunotherapy [22]. PD-L1 inhibitors, also known as checkpoint inhibitors, are a class of immunotherapy drugs used in the treatment of NSCLC. Recent publications have elucidated a subtle link between the expression status of PD-L1 and/or the clinical response of the inhibitor of this protein and human gut microbiome compositional changes in cancer [23,24]. Furthermore, fecal microbiome transplantation has been shown to restore the response to immune checkpoint inhibitor therapy in patients with melanoma [25,26].

The tumor microenvironment, pro-inflammatory factors, and immune checkpoints play a significant role in lung cancer development, progression, and treatment response [27,28]. As of today, we lack complete knowledge of how the gut and lung microbiomes interact with the immune system in tumor microenvironments. Furthermore, only a small number of studies have investigated the association between the lung tissue microbiome and clinical data of patients with lung cancer.

In this study, we performed a comprehensive analysis of the microbiome compositions of lung tumor and lung parenchyma tissues, alongside fecal samples obtained from individuals diagnosed with NSCLC. We aimed to determine the link between the bacterial compositions of NSCLC patients along the gut–lung axis in relation to clinical parameters, molecula r markers of systemic inflammation such as CRP (C-reactive protein), the NLR (neutrophil-to-lymphocyte ratio), LPS (lipopolysaccharide), CD8+ cells (cytotoxic T lymphocytes), and the expression levels of PD-L1 protein (Figure 1).

## 2. Results

### 2.1. Global Bacterial Profile Changes at Both Lung and Gut Levels in NSCLC Patients

Bacterial profile analysis showed that only some of the detected amplicon sequence variants (ASVs) (28%) were common between the lung tumor and lung parenchyma samples (Figure 2A). The same group of detected ASVs were common between NSCLC and control individuals’ stool samples (Figure 2B). Principal coordinate analysis (PCoA) supported the distinction of bacterial communities between the patient and control study groups using tissue samples (PC1—24.7% and PC2—7.3%) (Figure 2C), while the PCoA clusters of the stool samples were not so distinct between the two groups (PC1—5.4% and PC2—4%, Figure 2D). These results indicate that biopsies better represented the bacterial division between the NSCLC patients and the control groups.

The Ward.D2 hierarchical dendrogram shows that biopsy samples of NSCLC patients were clustered in two groups based on their microbiome compositions. The first cluster was mostly composed of lung tumor samples (79% lung tumor tissue and 21% parenchyma tissue), whereas the second was mostly composed of lung parenchyma samples (65% parenchyma tissue and 35% lung tumor tissue) (Figure 2E). No clustering was observed between patient and control group stool samples (Figure 2F). The obtained results further confirmed that the bacterial profiles of biopsies reflect the separation between tumorous and healthy lung tissue.

A permutational multivariate analysis of variance (PERMANOVA) demonstrated significant differences in microbiome composition between NSCLC patients’ tumor and parenchyma biopsy samples at all taxonomic levels (Appendix A). Significant differences were also observed between patient and control group stool samples. Moreover, the difference in microbiome composition between the analyzed groups was 8 to 15 times higher (F-values) in the lung tissue compared to that in the gut (stool). Therefore, these results emphasize that in NSCLC patients, the lung tissue microbiome is more sensitive for study group separation. However, we also observed that the global bacterial profile of the intestines of the patients differed from that of the control group, thereby confirming the connection between the lungs and gut.

### 2.2. Bacterial α-Diversity and Compositional Changes in NSCLC Patients at Lung and Gut Levels

An analysis of α-diversity revealed that the NSCLC tumor tissue samples had higher bacterial richness, diversity, and evenness compared to the lung parenchyma tissue samples (Figure 3A–D). No statistically significant differences in α-diversity were found between the NSCLC and control group stool samples (Figure 3E–H), which is in agreement with previously described outcomes. Although the global stool bacterial profiles differed between the cases and controls, the differences were not reflected in the global bacterial richness or diversity, suggesting that the differences in stool microbiome composition were most likely due to changes in the variations of individual bacteria.

The most abundant phyla in the lung tissue were Proteobacteria, Firmicutes, Bacteroidota, and Actinobacteriota. The top five genera were *Pseudomonas*, *Streptococcus*, *Veillonela*, *Actinomyces*, and *Prevotella* (Figure 4A–C). Differential analysis of bacterial abundance in lung tumor and lung parenchyma tissue revealed 56 differences at all taxonomic levels (Appendix A). Firmicutes and Bacteroidota were more abundant in lung tumor tissue samples compared to parenchyma tissue, whereas Proteobacteria were depleted in lung tumor tissue samples. At the genus level, the abundance of *Prevotella 7*, *Leptotrichia*, *Alloprevotella*, and *Cutibacterium* was increased in the lung tumor tissue samples, whereas *Pseudomonas* was depleted in the lung tumor tissue samples (Figure 5A and Appendix A).

The stool samples of both the NSCLC patients and the healthy control group were enriched with the Firmicutes, Bacteroidota, Actinobacteriota, Proteobacteria, and Verrucomicrobiota phyla. The most common bacteria at the genus level were *Bacteroides*, *Faecalibacterium*, *Collinsella*, *Blautia*, the *Christensenellaceae* R-7 group, *Holdemanella*, and *Ruminococcus* (Figure 4D–F). Differential analysis of the stool microbiomes between the NSCLC patients and healthy controls revealed 17 significantly abundant bacteria (Appendix A). Bacteroidota, Desulfobacterota, *Bacteroides*, *Alistipes*, and *CAG-873* were more abundant in NSCLC patients’ stool samples, while *Blautia*, *Romboutsia*, *Clostridium sensu stricto* 1, *Alistipes*, and *Fusicatenibacter* were more abundant in controls (Figure 5B and Appendix A).

### 2.3. Evaluation of Lung–Gut Axis at the Microbiome Compositional Level

An analysis indicated that the degree of shared bacteria (ASV and genus level) between the NSCLC patients’ lung biopsies and the NSCLC patients’ stool samples was significantly greater in cases involving tumor biopsy samples compared to cases where lung parenchyma was utilized. The same tendency was observed when comparing the biopsy samples of the NSCLC patients (tumor or parenchyma) to the control group’s stool samples (Appendix A). Furthermore, β-diversity analysis revealed a significant positive correlation between *Romboutsia* and *Alistipes* at the genus level in the lung tumor biopsies and the NSCLC patients’ stool samples (Appendix A).

### 2.4. Differences in Bacteria Abundance in Relation to NSCLC Clinical Data

Further changes in bacterial α- and β-diversity in NSCLC patients’ lung tumor tissues, parenchyma tissues, and stool samples depending on the patients’ demographic characteristics (smoking status) and clinical data (COPD, histology type, stage of NSCLC, degree of histologic differentiation in cancer cells, and CD8+ T-lymphocyte count in malignant lung tissue) were identified (Appendix A).

The lung tumor tissue of smokers was enriched with *Alloprevotella tannerae* and *Streptococcus salivarius*, whereas their stool microbiome was enriched with *Ruminococcus* species, *Holdemanella*, *Sellimonas*, *Solobacterium*, and *Succinivibrio* (Appendix A).

The tumor samples of patients with COPD had a lower abundance of *Fusobacterium nucleatum* and *Selenomonas sputigena* species, whereas the stool samples of these patients had fewer *Bacteroides*, *Enterococcus*, and *Prevotella stercorea* (Appendix A).

The stool samples of NSCLC patients with adenocarcinoma, compared with samples from patients with squamous cell carcinoma, had higher bacterial richness and Simpson diversity (*p* = 0.027 and *p* = 0.034, respectively) (Figure 6). No differences in α-diversity were found at the lung tissue level. Bacterial composition analysis revealed *Dialister*, *Haemophilus*, *Ligilactobacillus*, *Megasphaera*, *Veillonella*, and *Butyrivibrio crossotus* to be more abundant in the stool samples of NSCLC patients with adenocarcinoma (Appendix A). Tissue-level analysis revealed enrichment of *Tropheryma whipplei* in lung parenchyma tissues of patients with adenocarcinoma, and enrichment of *Neisseria subflava* in tumor tissues of patients with squamous cell carcinoma.

A lower degree of histologic differentiation in cancer cells was associated with enrichment of *Coriobacteriia* and the *Ruminococcus gnavus group* in stool samples of NSCLC patients (Appendix A).

An analysis of the histologic type of the tumor tissue revealed an increased abundance of *Streptococcus* and *Actinomyces* in stage I–II NSCLC patients’ tumor biopsies compared with those of stage III NSCLC patients (Appendix A). Meanwhile, *Neisseria* were more abundant in advanced cancer tumors, as well as in parenchyma tissue. The stool samples of patients with higher NSCLC stages had a higher abundance of *Escherichia-Shigella*, the *Lachnospiraceae NK4A136* group, *Succinivibrio*, and *Klebsiella pneumoniae*.

Immunohistochemical reactions with a monoclonal rabbit anti-human CD8 antibody for cytotoxic T lymphocytes were performed in 74 histologically verified NSCLC samples. A median of 78 (interquartile range: 109.67 cells) cytotoxic T lymphocytes were detected in the microenvironment. An analysis of the relationship between the NSCLC CD8 cell count in cancerous lung tissue and the microbiota composition revealed the enrichment of tumor tissue with *Burkholderiales*, *Christensenellales*, *Lachnospirales*, and *Staphylococcales* at the order level in NSCLC patients with CD8 cell counts above the median (Appendix A).

### 2.5. Gut Bacterial Diversity Is Associated with Survival Rate of NSCLC Patients

Survival analysis showed that NSCLC patients with Simpson’s diversity values (determined in stool samples) above the median had prolonged overall survival (OS) (*p* = 0.0426) (Figure 7). The Cox regression analysis hazard ratio (HR) was 2.06 (95% CI: 1.025–4.17), suggesting that NSCLC patients with lower bacterial diversity had significantly shorter OS. The median survival for a group of patients with higher bacterial diversity was 15 months, while for a group of patients with lower bacterial diversity, it was just over half as long, at 8 months. The largest differences in OS were observed in months 5 to 10. After 10 months from diagnosis, 71% (20 out of 28) of the patients with higher bacterial diversity were still alive, while in the group of patients with lower bacterial diversity, only 44% survived (12 out of 27 patients).

### 2.6. Association of Systemic Inflammation Markers LPS, NLR, and CRP with Microbiota Changes

To assess the relationship between members of the tissue and stool microbiome and systemic inflammation, we measured the levels of lipopolysaccharide (LPS). The LPS concentration was evaluated in 61 patients with NSCLC. The median concentration of LPS was 12.1 pg/mL (interquartile range: 20.2 pg/mL). A higher bacterial richness in the stool samples was associated with higher LPS levels in circulation (Figure 8). Interestingly, the stool samples of NSCLC patients with LPS concentrations below the median were enriched with *Hungatella*. Bacterial composition analysis revealed that *Corynebacterium* and *Streptococcus salivarius* were more abundant in the tumor tissue samples of NSCLC patients with higher LPS levels, whereas *Mesorhizobium* were more abundant in their parenchyma tissue samples (Appendix A).

Associations between lung and gut microbiota changes, the CRP level, and the NLR were not found in our study.

### 2.7. PD-L1 Expression in Malignant Lung Tissue

Immunohistochemical analysis of PD-L1 protein expression was performed in 70 histologically verified NSCLC patients’ tissue samples. The detected median PD-L1 expression level in the microenvironment of NSCLC was 5 (interquartile range: 69 levels).

The α-diversity analysis showed that the bacterial richness in the lung tumor tissue gradually decreased with an increase in the level of PD-L1 expression (Figure 9). β-diversity analysis revealed the enrichment of *Neisseria* and *Streptococcus infantis* in the tumor samples of patients with lower PD-L1 expression (Appendix A). Meanwhile, the parenchyma biopsies of NSCLC patients with higher PD-L1 expression were enriched with *Actinomyces graevenitzii* as well as *Streptococcus mitis*, and had decreased levels of *Porphyromonas gingivalis* as well as *Prevotella nanceiensis.* The stool samples of NSCLC patients with higher PD-L1 expression were enriched with *Erysipelotrichaceae* UCG-006, *Olsenella*, *Parasutterella*, the Rikenellaceae RC9 gut group, and *Klebsiella pneumoniae*. Meanwhile, such bacteria as *Catenisphaera*, *Eggerthella*, *Pseudomonas*, *Sellimonas*, and *Streptococcus* were more abundant in the stool samples of patients with lower PD-L1 expression.

## 3. Discussion

In this study, we performed a comprehensive microbiome analysis at the gut and lung levels in a large group of well-characterized NSCLC patients. We determined the microbiome compositions in lung tumor and parenchyma biopsies as well as stool samples, and analyzed them in relation to clinical and molecular markers in the NSCLC patients.

Previous studies revealed that the intestinal and lung microbiome in NSCLC patients is characterized by a loss of microbial diversity [29,30]. Meanwhile, our study’s findings are in line with recent reports where α-diversity was higher in NSCLC patient samples, including tumor tissue, bronchoalveolar lavage (BALF), and saliva samples, compared with the control groups [31,32,33]. The relationship between increased alpha diversity and lung cancer development remains unknown. Interestingly, a recent meta-analysis demonstrated longitudinal variability in stool microbiome α-diversity in patients with different cancers across many studies [34]. In line with some previous reports [35,36], we found no significant differences in α-diversity when comparing stool samples between the NSCLC patients and the control group.

Our study showed significant differences at all taxonomic levels in the β-diversity of NSCLC patients’ lung tumor and parenchyma biopsy samples, supporting the findings of previous studies [29,37,38]. In accordance with our results, a study by Zeng et al. showed that BALF samples from NSCLC patients were enriched with *Streptococcus*, *Prevotella*, *Veillonella*, *Neisseria*, *Actinomyces*, *Alloprevotella*, and *Porphyromonas* [33]. These results suggest that BALF and lung tumor samples from NSCLC patients could yield similar microbial profiles. These bacteria may be associated with the upregulation of the extracellular signal-regulated kinase (ERK) and phosphoinositide 3-kinase (PI3K) signaling pathways in airway epithelial cells, leading to lung cancer development [39].

Several studies noted that the stool microbiome composition differed between NSCLC patients and healthy volunteers [40,41]. Similar to Zheng et al., we found that Bacteroidota, including beneficial bacteria of *Bacteroides*, were more abundant in NSCLC patients’ stool samples compared to the control group [40]. Lung cancer may be associated with decreases in beneficial bacteria and increases in pathogenic microorganisms, which also belong to the Bacteroidota phylum. Meanwhile, in our study, the *Blautia* genus belonging to the Firmicutes phylum was decreased in NSCLC patients’ stool samples, as in previous studies [36,41]. *Blautia* has beneficial anti-inflammatory and antitumor effects due to increased butyrate production, which may induce the expression of a central transcription factor (T-bet) and interferon-gamma (IFN-γ) in T lymphocytes [42,43,44]. These findings suggest that the depletion of *Blautia* may be crucial for cancer development.

Previous studies indicated that NSCLC patients’ gut–lung microbiome composition varies according to aging; smoking; COPD; the histological type, stage, and grade of lung cancer; and the PD-L1 expression levels on tumor cells [16,32,40]. Similar to another study, we found that *Neisseria subflava* was enriched in lung tumor tissues with squamous cell carcinoma compared with adenocarcinoma [45]. Several studies noted that *N. subflava* was the predominant bacteria in oral cavity squamous cell carcinoma patients’ samples [46]. We suggest that microaspirations of *Neisseria* species may also be associated with squamous cell carcinoma development in the lungs. Moreover, *N. subflava* may weaken the integrity of the mucus barrier, induce chronic inflammation, and cause distinct gene expression changes in murine lungs, which could promote carcinogenesis [47,48]. Interestingly, the presentation of *Tropheryma whipplei* in human lungs is very rare, but is commonly associated with immunosuppressed patients [49]. This may be the first microbiome study to find that *T. whipplei* was enriched in lung parenchyma biopsy samples from NSCLC patients with lung adenocarcinoma. *T. whipplei* may increase IL-10 and transforming growth factor beta (TGF-β) expression and decrease IFN-γ expression. As a result, immune system cells cannot present antigens or phagocytose bacteria [50,51].

We found that *Streptococcus* and *Actinomyces* were more abundant in early-stage tissues, and *Neisseria* was more abundant in advanced NSCLC patients’ tumor tissues. *Neisseria* species may modify host cell integrity and a signaling cascade, triggering cellular malignancies [52,53]. *Streptococcus* species, including *S. pneumoniae* and *S. pyogenes*, are linked to lung cancer development via chronic lower respiratory tract infections and dysbiosis [54]; microaspirations of resident oral cavity bacteria, such as *S. oralis* and *S. mutans*; and the induction of interleukin production and T-cell activation [55,56]. Moreover, *S. mutans* infections in the oral cavity are associated with chronic inflammation, IL-6 production, and epithelial transformation, which may cause cancer development and locally advanced disease stages [54,57]. This evidence provides a strong background, suggesting that *Streptococcus* may be associated with lung cancer initiation. Meanwhile, *Neisseria* may be related to lung cancer progression.

In line with recent studies, we noted that *Alloprevotella tannerae* and *Streptococcus salivarius* were enriched in lung tumor tissue samples from NSCLC patients who were smokers [58,59]. The bacteria *Alloprevotella tannerae* may promote oral inflammatory processes and increase the risk of oral squamous cell carcinoma due to increased nitrate intake while smoking and poor oral hygiene [60]. Interestingly, *Streptococcus salivarius* is a predominant commensal human oral cavity bacterium that may inhibit inflammatory pathways, modulate human epithelial cell immune responses, prevent colonization by pathogenic bacteria such as *S. pyogenes* and *S. pneumoniae*, and may have an antitumor effect on cells [61,62]. We suggest that microaspirations of *Alloprevotella tannerae* may induce NSCLC development, while microaspirations of *S. salivarius* may have protective properties in the lungs as well as in the oral cavity.

LPS is an endotoxin that arises from the outer membranes of Gram-negative bacteria. Even low concentrations of LPS may signal chronic inflammatory or metabolic diseases [63]. Our study’s results showed that a higher LPS concentration was associated with increased bacterial richness in NSCLC patients’ stool samples. Also, NSCLC patients’ parenchyma samples were enriched with the Gram-negative bacteria *Mesorhizobium.* The same bacteria were found in renal carcinoma, ovarian carcinoma tissue, colorectal cancer tissues, and BALF samples from NSCLC patients. Furthermore, recent studies showed that LPS activates inflammasomes in cancer cells and could prolong cancer cell survival [64,65,66,67,68].

PD-L1 is a trans-membrane protein and one of several immune evasion mechanisms. It is also a positive predictor and negative prognostic biomarker for NSCLC patients [69,70]. Our study may be the first to show associations between the PD-L1 expression level and α- and β-diversity in lung tumors, parenchyma biopsies, and stool samples. Studies using mouse tumor models showed that α-diversity could increase the production of pro-inflammatory cytokines and significantly improve tumor control upon anti-PD-L1 treatment [71]. However, the interplay mechanisms of α-diversity and the PD-L1 expression level in humans remain unknown. Next, we found that *Neisseria* and *Streptococcus infantis* were enriched in lung tumor biopsy samples with lower PD-L1 expression, similar to a previous study [24]. Furthermore, *S. infantis* and *S. mitis* strains are highly similar and belong to the same commensal species *Mitis* group [72]. *S. mitis* may modulate PD-L1 expression and IL-10 levels through prostaglandin E2 (PGE2) in monocytes. A lower level of PGE2 production was associated with lower PD-L1 expression in monocytes [73]. We suggest that *S. infantis* may have properties similar to *S. mitis* and may modulate PD-L1 expression on the cell surface. A recent study noted that *Neisseria gonorrhea* increases IL-10 secretion and leads to the upregulation of PD-L1 in human cells such as macrophages [74]. We suppose that *Neisseria* could modulate PD-L1 expression on tumor cells, similar to their effect on macrophages. However, the current relationship between these correlations is unknown, and further investigation is needed.

Our study, for the first time, showed a significant association between survival data from NSCLC patients with early- to advanced-stage disease and α-diversity, while other authors only investigated patients with advanced disease [27,75,76]. We found that higher bacterial richness was associated with prolonged survival. Meanwhile, other studies suggested that a higher α-diversity in the stool microbiome could be associated with higher PD-L1 expression levels, better responses, and prolonged progression-free survival in NSCLC patients treated with immunotherapy [75,77].

The gut–lung axis is a complex system that connects, changes, and affects the microbiome from the gastrointestinal tract to the lungs [78]. The bacterial products of the gut microbiome may cross the epithelial barrier into the bloodstream and regulate the gut–lung axis [79]. We found a significant positive correlation between *Romboutsia* and *Alistipes* species along the gut–lung axis in NSCLC patients. The current relationship between these correlations remains unknown, and further investigation is needed. However, other microbiota studies noted that *Alistipes* predominate colorectal cancer mucosa compared with healthy controls, may cause elevated IL-6 levels in the blood, and may promote cancer development by inducing cell growth [80,81]. Meanwhile, colorectal cancer studies noted that *Romboutsia* is decreased in cancer mucosa, and could be associated with a benign tumor or early tumor formation [82]. Furthermore, *Romboutsia* is generally identified in healthy human stool samples, and may be used as a potential microbial indicator of a disease condition [83].

Our study had a few limitations that need to be acknowledged. This study proceeded during the COVID-19 pandemic, which also impacted a limited number of participants in our research. We could not evaluate or compare the response to immunotherapy or the interaction between the gut and the lung microbiome due to the low number of patients who were treated with immune checkpoint inhibitors in this cohort of patients. Also, this study requires validation through research involving lung cancer patient cohorts from various geographical locations, which would encompass diverse nutritional habits and ethnic groups, to ensure the generalizability and applicability of the findings across different populations. Finally, the study results derived from next-generation sequencing technologies should be cross-verified with traditional culture-based methods. Further extensive randomized clinical trials are required to validate our findings.

In conclusion, we demonstrated that the lung microbiome and the stool microbiome are separate bacterial milieux in patients with NSCLC. Significant associations between the host microbiome and smoking habits, histology type, tumor differentiation degree, CD8+ cells in the tumor microenvironment, and the bacterial endotoxin LPS were observed. Survival analysis revealed that higher bacterial richness was associated with prolonged survival in NSCLC patients. We also showed that the PD-L1 expression in tumor tissue is linked to bacterial richness and specific bacterial species, suggesting important implications for NSCLC patients who are treated with checkpoint inhibitor therapy.

## 4. Materials and Methods

### 4.1. Study Cohort

This prospective study was performed at the Hospital of the Lithuanian University of Health Sciences Kauno Klinikos, Department of Pulmonology, and the Institute for Digestive Research, from 2019 to 2023. This study was approved by the Kaunas Regional Biomedical Research Ethics Committee (Protocol No. BE-2-51, Lithuania, Kaunas) and registered in the United States National Institutes of Health trial registry at ClinicalTrials.gov with the identifier NCT05164445. All study subjects signed an informed consent form to take part in this study.

A total of 200 patients were referred to our tertiary university hospital with suspected LC on a chest computed tomography (CT) scan. LC patients were excluded if they had a history of another malignancy, active infections, or had used antibiotics in the past six months. In total, we enrolled 105 patients in our study after histological NSCLC confirmation. Additionally, 54 healthy volunteers without chronic diseases, malignancy, or antibiotic use in the last six months were included in the control group. For microbiome analysis, we obtained lung tumor tissue, lung parenchyma tissue, and stool samples from the NSCLC patients. Peripheral blood samples were also collected from the NSCLC patients. The control group only donated stool samples.

### 4.2. Biopsy and Stool Sample Collection

Lung tumor biopsy samples were collected as described previously [84]. The lung tumor biopsy samples were placed in a 10% neutral buffered formalin solution for histological examination or a sterile cryotube for microbiome evaluation (stored at −80 °C until processing). Lung parenchyma biopsies were collected using a flexible fibro bronchoscope (BF-H190, Olympus Evis Exera III, Olympus, Tokyo, Japan) under fluoroscopy (Philips BV Pulsera Fluoroscopy C-arm, Tokyo, Japan) control with single-use forceps (Olympus, Tokyo, Japan). During the procedure, we took five lung parenchyma biopsy samples using new single-use forceps. To ensure the collection of a parenchymal biopsy from a non-malignant section of the lung, the biopsies were taken from different bronchi. Furthermore, fluoroscopy was utilized to confirm that the forceps were positioned more than 5 cm away from the lung tumor site.

Stool samples were collected from each participant using sterile stool sample collection kits. All stool samples were collected at the Hospital of LUHS Kauno Klinikos, Pulmonology Department. After collection, the stool samples were immediately transferred to the laboratory and placed in sterile cryotubes that were frozen at −80 °C until future use.

### 4.3. Pathological Examination

The NSCLC tissue samples were fixed in a 10% formalin solution and embedded in paraffin. Tissue sections 3–5 μm thick were cut and subsequently de-waxed and rehydrated through graded alcohols. The expression of PD-L1 and the number of CD8+ T cells in the paraffin-embedded lung tumor tissue samples were analyzed via immunohistochemistry using a Ventana BenchMark XT staining machine (Ventana Medical Systems, Roche, France). Primary antibody detection was performed using commercially available kits following the manufacturer’s instructions. Monoclonal rabbit anti-human antibodies were used for the identification of CD8+ T cells (clone C8/144B) and PD-L1 expression (SP263, Ventana). A quantitative evaluation of PD-L1 expression was visualized in the three most representative high-power fields (HPFs, 400× magnification) per tissue section, using an Olympus BX50 microscope (Olympus Co, Tokyo, Japan). The number of cells with positive staining was counted manually. A CD8+ T-cell evaluation was performed using a “DAKO EnVision Flex” (DAKO, K8000 Denmark) visualization system by counterstaining sections with Mayer’s hematoxylin. Microsections were scanned with “Pannoramic Viewer” (3D HISTECH Ltd., RRID: SCR_014424, Budapest, Hungary), and in each case cytotoxic T lymphocytes were counted with a marker counter in three annotated 304 558 µm^2^ microscopic fields in the “Slider Viewer” (3D HISTECH Ltd., RRID: SCR_014424, Budapest, Hungary) digital microimaging analysis program. Statistical analysis was carried out using the Mann–Whitney U test and the Kruskal–Wallis test with pairwise comparisons (*p* < 0.05).

### 4.4. Blood Sample Collection and Biomarker Testing

Peripheral blood samples were collected from the NSCLC patients in BD Vacutainer^®^ (BD Bioscience, San Jose, CA, USA) serum tubes. After the collection of the blood, the samples were allowed to clot for one hour at room temperature (23–24 °C) and, after that time, were centrifuged at 4000 rpm for 10 min. Following centrifugation, the serum was separated from the blood and transferred into sterile 1.0 mL cryotubes using a Pasteur pipette. The lipopolysaccharide (LPS) concentrations in the serum were analyzed using commercially available enzyme-linked immunosorbent assay kits (LPS ELISA kit, USACSB-E09945h, CUSABIO, Wuhan, China) according to the manufacturer’s instructions. All samples were analyzed in duplicate.

### 4.5. DNA Extraction and Library Preparation

Extraction of bacterial DNA from all collected lung biopsies (tumor and parenchyma) and stool samples was performed using a commercially available microbiome DNA purification kit (Thermo Fisher Scientific, Waltham, MA, USA), following the manufacturer’s recommendations. PCR 27F and 338R universal 16SrRNA gene primers were used for V1–V2 region amplification. The amplification of the appropriate fragment size after PCR was verified using electrophoresis. For PCR product normalization, a 96-well SequalPrep normalization plate kit (Thermo Fisher Scientific, Waltham, MA, USA) was used. A Collibri nucleic acids library prep kit (Thermo Fisher Scientific, Waltham, MA, USA) was used for final library concentration optimization. The MiSeq platform was used for 16S rRNA V1–V2 region sequencing (2 × 300 bp, Illumina, Hayward, CA, USA). Water control samples on the stage during DNA extraction and library preparation were included to evaluate possible contamination.

### 4.6. Bioinformatic and Statistical Data Analysis

Bioinformatic and statistical analyses were performed in the R software (version 4.1.2) environment as described previously [85]. Briefly, the dada2 package (version 1.16.0) was used to prepare pre-processed paired-end FastQ files containing no barcodes or adapters. The samples were denoised, quality checked, and annotated for taxonomic affiliation using a naive Bayesian classification based on the SILVA bacterial database (version 138). Bacterial sequences called amplicon sequence variants (ASVs) and a table containing counts, taxonomy, and clinical data were used for further analysis. Using prepared water control samples and a Decontam R package (version 4.1.2), a presumed ASV contaminant was removed from the analysis. Bacteria at the phylum level with counts less than 50 were removed from the analysis, as well as bacteria with no taxonomic affiliation.

To evaluate the bacterial community α-diversity, all of the samples were resampled to an equal library size using the phyloseq package (version 4.1.2). An analysis of α-diversity was performed at the ASV level. Bacterial richness, Pielou’s evenness, and Simpson’s and Shannon’s diversity indices were calculated using the vegan package (version 4.1.2). To reveal statistically significant differences between study groups, the Mann–Whitney test or the Kruskal–Wallis test was carried out, followed by pairwise Wilcoxon rank sum tests. For multiple comparisons, Benjamini–Hochberg FDR correction was used.

Principal coordinate analysis (PCoA) and the Ward.D2 algorithm with a Bray–Curtis distance matrix were used to visualize the data, obtain a set of principal coordinates, and show the clustering of the samples. Differences between study groups at each taxonomic level were evaluated via a permutational multivariate analysis of variance (PERMANOVA) (9999 permutations). Data normalization and β-diversity analysis were performed using the DESeq2 package (version 4.1.2). β-diversity analysis was performed only on bacteria that accounted for more than 25% of the samples in at least one of the compared groups. The survival and survminer packages (version 4.1.2) were used for survival analysis, using the Cox method to evaluate differences between survival curves. For all tests, differences were considered significant when the *p*-value corrected for multiple comparisons (*p* adj.) was <0.05.

## 5. Conclusions

This study demonstrated that the lung microbiome and the stool microbiome are separate bacterial milieux in patients with NSCLC. Significant associations between the host microbiome and smoking habits, histology type, tumor differentiation degree, CD8+ cells in the tumor microenvironment, and the bacterial endotoxin LPS were observed. Survival analysis revealed that higher bacterial richness was associated with prolonged survival in NSCLC patients. We also showed that the PD-L1 expression in tumor tissue is linked to bacterial richness and specific bacterial species, suggesting important implications for NSCLC patients who are treated with checkpoint inhibitor therapy.

## Figures and Tables

**Figure 1 ijms-25-02323-f001:**
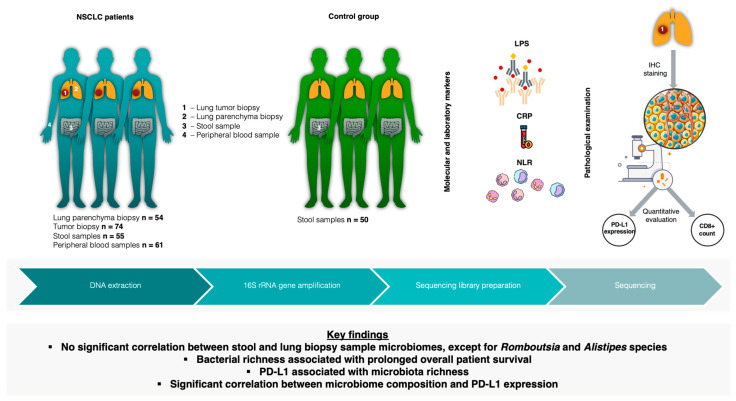
Study design illustration. n—number of patients; NSCLC—non-small-cell lung cancer; LPS—lipopolysaccharide; CRP—C-reactive protein; NLR—neutrophil-to-lymphocyte ratio; IHC—immunohistochemistry; PD-L1—programmed death ligand 1.

**Figure 2 ijms-25-02323-f002:**
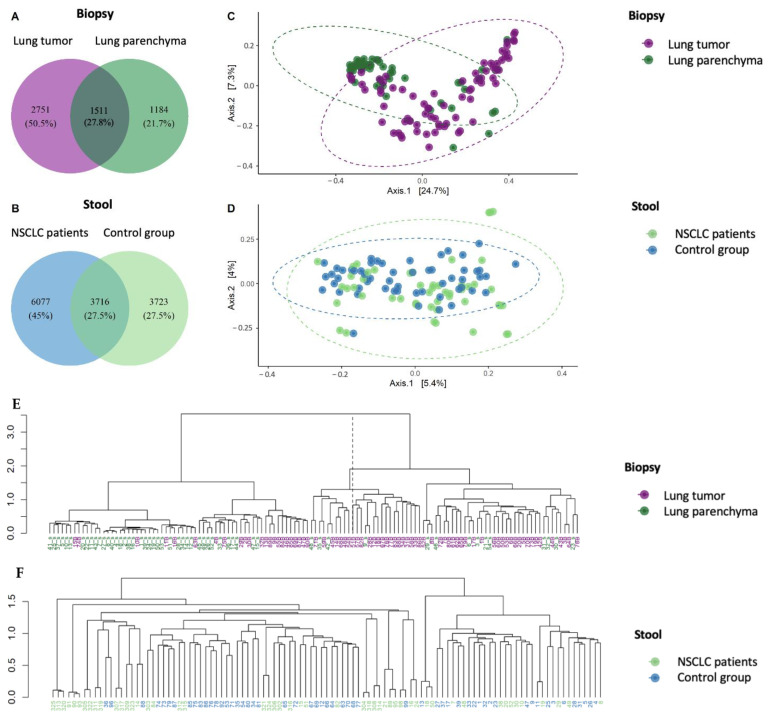
The bacterial compositions of lung biopsy and stool sample microbiomes of NSCLC and control group patients. The Venn diagrams indicate the numbers of group-specific amplicon sequence variants (ASVs) between lung tumor and lung parenchyma biopsy samples (**A**) and between NSCLC and control group stool samples (**B**). Principal coordinate analysis (PCoA) of biopsy (**C**) and stool samples (**D**). Ward.D2 hierarchical clustering of biopsy (**E**) and stool samples (**F**).

**Figure 3 ijms-25-02323-f003:**
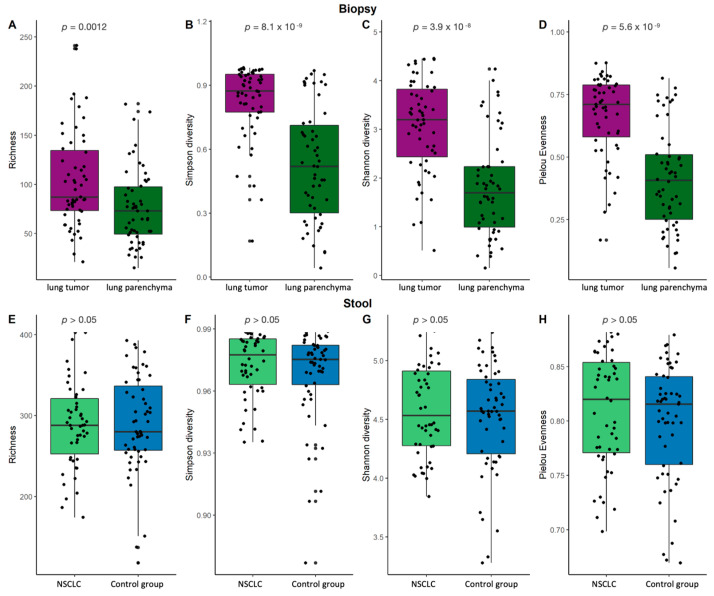
Lung biopsy and stool sample α-diversity comparison. Bacterial richness, Simpson’s diversity, Shannon’s diversity, and Pielou’s evenness diversity for biopsy (**A**–**D**) and stool (**E**–**H**) samples. *p*—the significance level.

**Figure 4 ijms-25-02323-f004:**
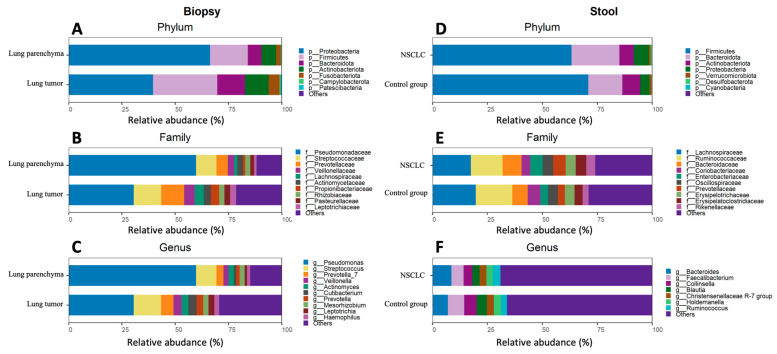
Lung biopsy and stool sample bacterial composition plots of NSCLC patients and control group. Relative abundance of the top 7 phyla, top 10 families, and top 10 genera for the biopsy (**A**–**C**) and stool (**D**–**F**) samples.

**Figure 5 ijms-25-02323-f005:**
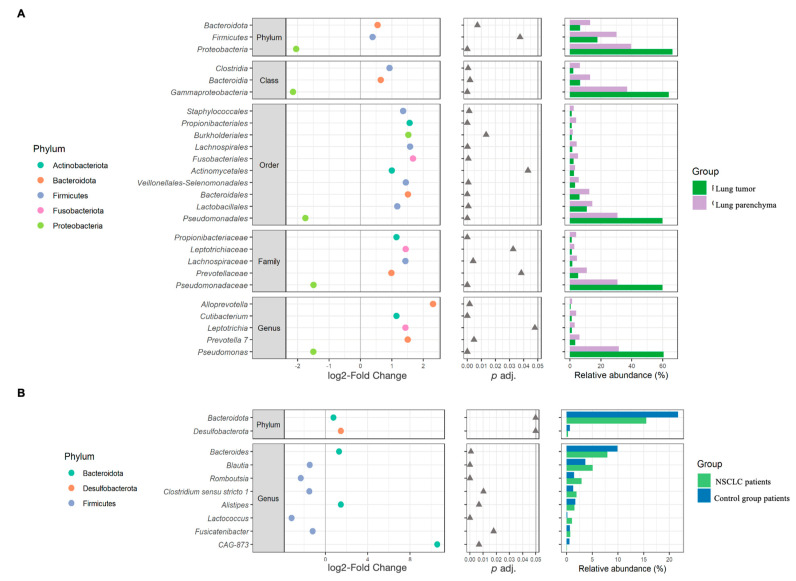
β-diversity analysis results. The left plots represent the log2-fold change values of the mean normalized abundance ratios between the comparison groups. The middle plots represent the *p* adj. values obtained after differential abundance analysis. The right plots represent the mean relative abundance values. This figure presents only those significantly different bacteria between comparison groups that accounted for at least 1% of the total number of taxa in at least one study group (the whole list of statistically significant results is presented in Appendix A). (**A**)—biopsies; (**B**)—stool.

**Figure 6 ijms-25-02323-f006:**
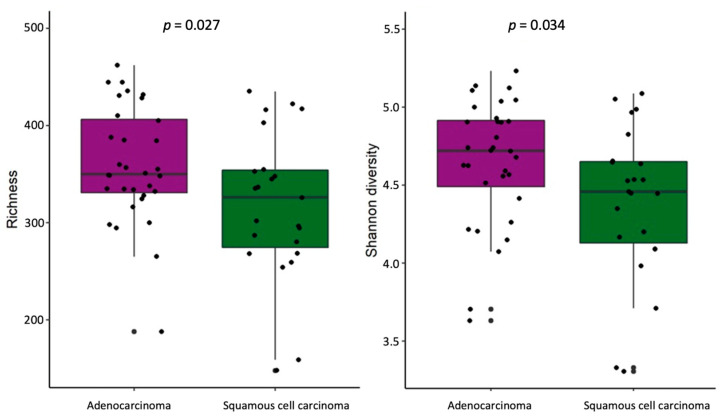
α-Diversity comparison according to histology type in NSCLC patients’ stool samples. The bacterial richness and Shannon’s diversity of NSCLC patients’ stool samples. *p*—the significance level.

**Figure 7 ijms-25-02323-f007:**
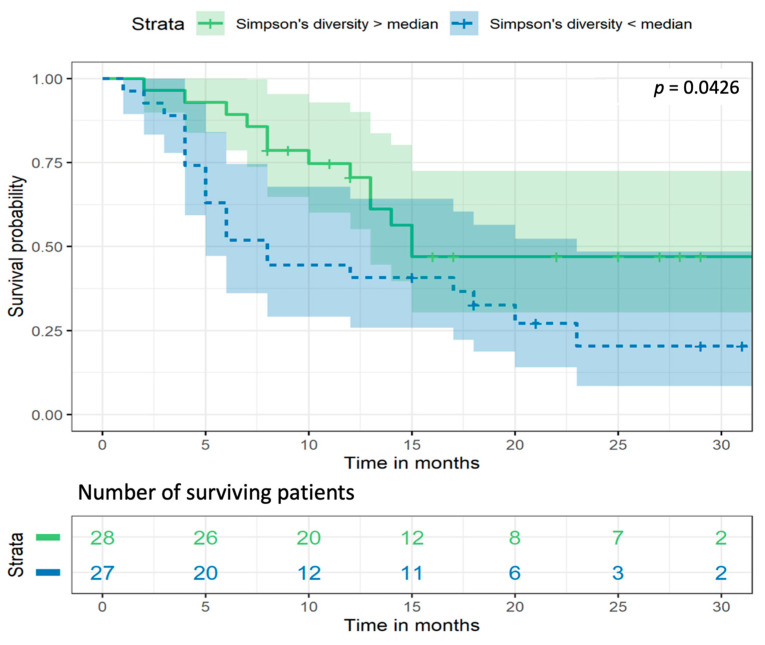
Kaplan–Meier survival analysis curves for Simpson’s bacterial diversity in NSCLC patients’ stool samples. *p*—the significance level.

**Figure 8 ijms-25-02323-f008:**
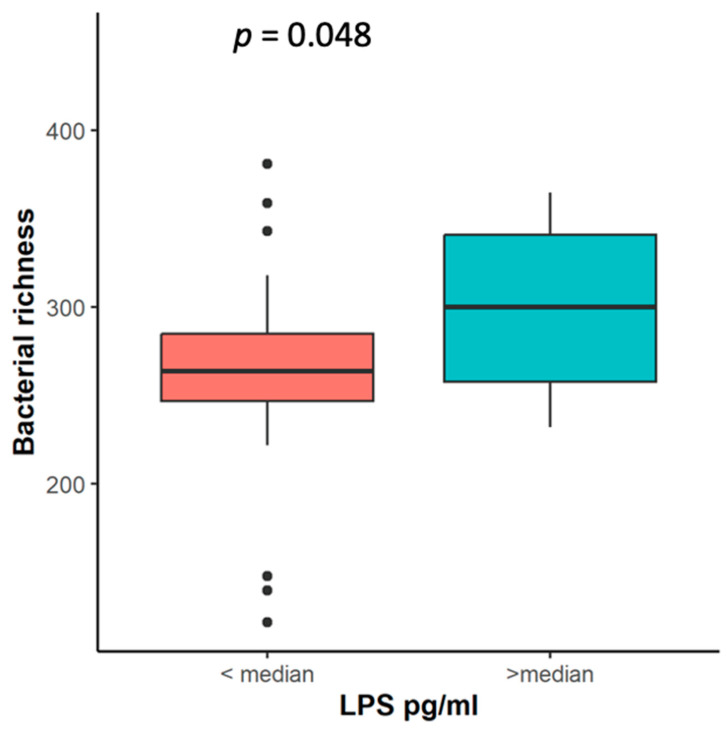
Bacterial richness in NSCLC patients’ stool samples according to LPS level. LPS—liposaccharide concentration (pg/mL), *p*—the significance level.

**Figure 9 ijms-25-02323-f009:**
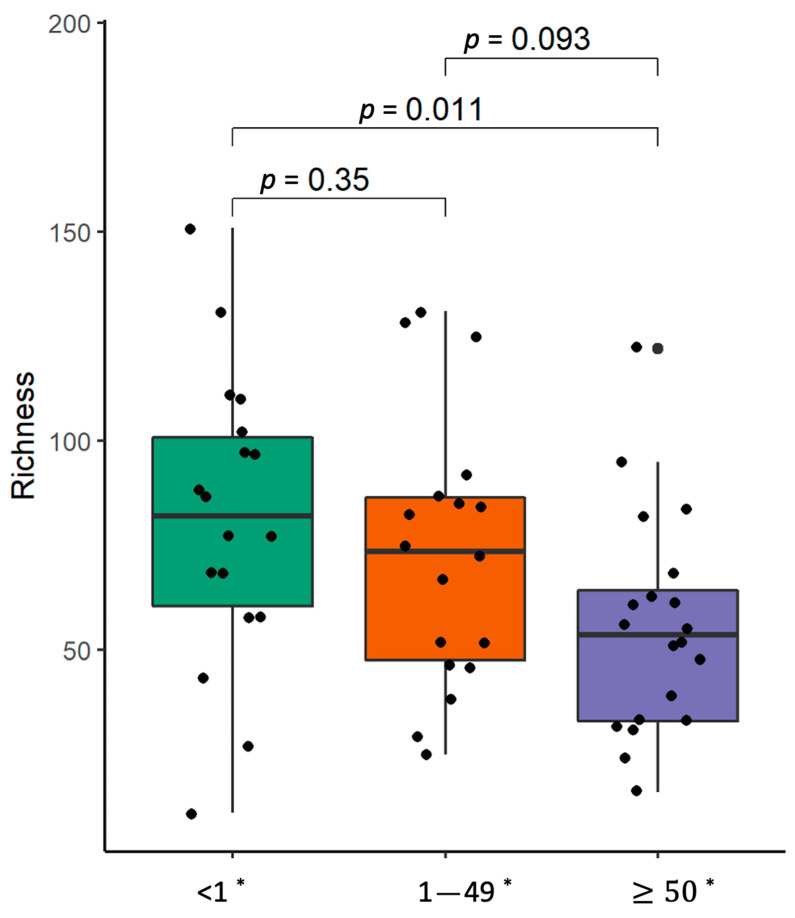
α-Diversity comparison according to PDL-1 expression level in non-small-cell carcinoma lung tissue samples. The bacterial richness of NSCLC patients’ samples. *—PD-L1 expression level, *p*—the significance level.

## Data Availability

The data presented in this study are available on request from the corresponding author.

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
