# Peer review of "Detailed Characterization of the Lung–Gut Microbiome Axis Reveals the Link between PD-L1 and the Microbiome in Non-Small-Cell Lung Cancer Patients"

_ijms, 2024, doi:10.3390/ijms25042323_

Round 1

Reviewer 1 Report

Comments and Suggestions for Authors

This is a very interesting manuscript by Dr. Ankudavicius and collaborators on the relationship between the stool and lung biopsy microbiome, lung cancer, expression of critical immune molecules and survival. 

Author list: please check the superscript markers.

line 71-77: ASV, and other abbreviations not explained. I would try to reduce the number of abbreviations used.

Legend to figure 2,A, B: do the numbers represent the numbers of reads analyzed.  E. Please explain colors (samples from controls tissue vs cancer tissue).

Line 117: Figure 5A, 3rd panel, it looks as if the tumor has a higher relative abundance of Pseudomonas than the control lung tissue (green bar vs violet bar).

Supplement Figure 1 and Supplement Figure 2 appear to be the same. Figure legend, please explain abbreviation.

Supplement Table 1: are the P values the real values, or calculated to a specific set point, e.g. p=0.001. I would provide the actual numbers.  Same for Supplement Table 2, 3, 5.

Supplement Table 1, 2, 3, 4, 5, 6, 7, 8, 9, 10, 11, 12 need Table legend, explanation what the rows are, and what the colors mean

Limitations of the study: I would mention that this type of study needs validation from cancer patient cohorts from other geographical locations representing different nutrition and ethic groups.

Author Response

Dear Reviewer,

Thank you for your remarks and suggestions, they improved a better quality of this article.

Author list: please check the superscript markers.

Revised and corrected.

line 71-77: ASV, and other abbreviations not explained. I would try to reduce the number of abbreviations used.

We had included explanations of such abbreviations as ASV, PERMANOVA, PCoA.

Legend to figure 2,A, B: do the numbers represent the numbers of reads analyzed.  E. Please explain colors (samples from controls tissue vs cancer tissue).

The number in the Venn diagram represents the number of different detected bacterial sequences (ASV) detected in those groups (uniquely or both in two groups). We had adjected an explanation to the figure description.  We also had adjusted the legends for the plot E and F in Figure 2 for color code explanation.

Line 117: Figure 5A, 3rd panel, it looks as if the tumor has a higher relative abundance of Pseudomonas than the control lung tissue (green bar vs violet bar).

The legend of the figure was corrected.

Supplement Figure 1 and Supplement Figure 2 appear to be the same. Figure legend, please explain abbreviation.

Corrected, the missing supplementary table was added.

Supplement Table 1: are the P values the real values, or calculated to a specific set point, e.g. p=0.001. I would provide the actual numbers.  Same for Supplement Table 2, 3, 5.

The p-values in Supplement Table 1 are derived from selected PERMANOVA permutations. As PERMANOVA offers an initial perspective, we deem it unnecessary to recompute the analysis. In other supplementary tables are presented real (full) p-values. To enhance table readability, numbers beyond three decimal places are hidden. However, readers can reveal the complete values by directly clicking on the cell in the Excel table.

Supplement Table 1, 2, 3, 4, 5, 6, 7, 8, 9, 10, 11, 12 need Table legend, explanation what the rows are, and what the colors mean

Different colors were used for easier data separation between groups but there is no need to leave it in the final version of supplementary tables. For this reason, we already removed the colors. The missing value in the first row of each table was entered. 

Limitations of the study: I would mention that this type of study needs validation from cancer patient cohorts from other geographical locations representing different nutrition and ethic groups.

Your suggestions were added as limitations in the line 393-397.

Sincerely,

Authors

Reviewer 2 Report

Comments and Suggestions for Authors

The abstract part need to rewrite, the main results should be provided in the abstract part,

In fig 1A, FIG3a-d, it seems the tumor has more bacteria than that in lung parenchyma, what is the reason? And do this trend similar with published papers?

Do authors laso using the culture method to culture bacteria to verify the sequencing results? As we all know, the results of genomics have significant uncertainty

The figs is not as a normal list, please adjust this

Why authors conduct the α-diversity with PDL-1 expression level? The α-diversity means a macro index, specific bacteria at species level or genus level has more practice meaning. In addition, authors using PCR 27F and 338R universal 16S rRNA gene primers were for V1–V2 region amplification, this technology can noy have the capability to the species level, so all results on bacteria at species level is unreliable.

Comments on the Quality of English Language

The abstract part need to rewrite, the main results should be provided in the abstract part,

In fig 1A, FIG3a-d, it seems the tumor has more bacteria than that in lung parenchyma, what is the reason? And do this trend similar with published papers?

Do authors laso using the culture method to culture bacteria to verify the sequencing results? As we all know, the results of genomics have significant uncertainty

The figs is not as a normal list, please adjust this

Why authors conduct the α-diversity with PDL-1 expression level? The α-diversity means a macro index, specific bacteria at species level or genus level has more practice meaning. In addition, authors using PCR 27F and 338R universal 16S rRNA gene primers were for V1–V2 region amplification, this technology can noy have the capability to the species level, so all results on bacteria at species level is unreliable.

Author Response

Dear Reviewer,

Thank you for your remarks and suggestions, they improved a better quality of this article.

The abstract part need to rewrite, the main results should be provided in the abstract part,

Corrected, the main result added in the abstract.

In fig 1A, FIG3a-d, it seems the tumor has more bacteria than that in lung parenchyma, what is the reason? And do this trend similar with published papers?

We discussed this trend in discussion. Also, we corrected this paragraph according to your comment in the line 269-270.

Do authors laso using the culture method to culture bacteria to verify the sequencing results? As we all know, the results of genomics have significant uncertainty

No, this was based only on the NGS technology, which offers a more rapid, comprehensive, and detailed analysis of genetic material compared to culture methods, making it a powerful tool for modern microbiological and genetic research. However, we had added to the discussion part the need for validation of the detected results (line 397-399).

The figs is not as a normal list, please adjust this

Revised and corrected.

Why authors conduct the α-diversity with PDL-1 expression level? The α-diversity means a macro index, specific bacteria at species level or genus level has more practice meaning. In addition, authors using PCR 27F and 338R universal 16S rRNA gene primers were for V1–V2 region amplification, this technology can noy have the capability to the species level, so all results on bacteria at species level is unreliable.

PD-L1 is a positive predictive and negative prognostic biomarker for NSCLC patients. Clinical studies of NSCLC patients showed that microbiome is a predictive biomarker for immunotherapy as well. However, a single biomarker may not be sufficient to predict optimal treatment responses and survival. For this reason, we try to investigate associations between PD-L1 and microbiome, including alpha and beta diversities. Our study revealed novel data and may be the first to show associations between  PD-L1 levels and alpha diversity in lung tumor samples. Also, associations between PD-L1 and microbiome composition are observed and presented in our study.

We concur that the 16S rRNA genes V1-V2 region alone is insufficient for conducting α-diversity analysis at the genus or species level. Therefore, in our study, we carried out the analysis at the ASV level, representing the most granular level of detection for bacterial sequences. We adjusted the methods part with this information in the line 497.

Comments on the Quality of English Language is the same. Additional answers are not provided.

Sincerely,

Authors

Reviewer 3 Report

Comments and Suggestions for Authors

 The submitted manuscript describes an important and currently hot topic of the connection of gut microbiota with human health. Here, the Authors investigated the gut-lung axis in patients with non-small cell lung cancer (NSCLC), and the association between microbiota of gut and lung (malignant and healthy tissues) and some clinical parameters (CRP, NLR, LPS, CD8, PD-L1). The methodology part of the manuscript is properly presented, the results section is clear and easy to follow, and the conclusions are supported by the results. I have only minor comments on the manuscript.

Minor remarks

Line 46 „including”

Line 70 „C-reactive”

Line 135 „Verrucomicrobiota”

Line 256 please explain “BALF” at the first use

Line 273, 275 “probiotic bacteria of Bacteroides” please, avoid using “probiotic” for a group (here a genera) of bacteria as this term is only for single strains that have positive, scientifically proven effects on human organism. Please use “beneficial” instead.

Line 288 “Neisseria species”

Line 327-329 please provide a reference

Line 373-38 please delete the repeating sentences

Line 462 „16S rRNA”

Line 467 “as described previously” please provide a reference

Author Response

Dear Reviewer,

Thank you very much for your remarks and suggestions, they improved the quality of this article.

Line 46 „including”

Corrected.

Line 70 „C-reactive”

Corrected.

Line 135 „Verrucomicrobiota”

Corrected.

Line 256 please explain “BALF” at the first use

Corrected.

Line 273, 275 “probiotic bacteria of Bacteroides” please, avoid using “probiotic” for a group (here a genera) of bacteria as this term is only for single strains that have positive, scientifically proven effects on human organism. Please use “beneficial” instead.

Corrected.

Line 288 “Neisseria species”

Corrected.

Line 327-329 please provide a reference

Corrected.

Line 373-38 please delete the repeating sentences

Deleted. 

Line 462 „16S rRNA”

Corrected.

Line 467 “as described previously” please provide a reference

Corrected.

Sincerely,

Authots

Reviewer 4 Report

Comments and Suggestions for Authors

Thank you for the invitation to review this interesting article. This is an article on the characterization of the lung microbiome in cells from patients with non-small cell lung cancer, specifically in the search for a link between PDL1 and the pulmonary microbiome. In general, the article is very well-founded, the results are described in detail, the graphical support seems quite good to me, and the discussion and conclusions are appropriate in the context of the results.

The only comment that I consider essential before its possible publication is the addition of the "Materials and methods" section, especially with the purpose of describing in detail the processing of the samples and the materials used in the sequencing.

Author Response

Dear Reviewer,

Thank you very much for your remarks and suggestions, they improved the quality of this article.

The only comment that I consider essential before its possible publication is the addition of the "Materials and methods" section, especially with the purpose of describing in detail the processing of the samples and the materials used in the sequencing.

The section was retitled to "Materials and methods" (lines 389-492). Also, additional information was written in this part of the article (the yellow highlighter).

Sincerely,

Authors

Round 2

Reviewer 2 Report

Comments and Suggestions for Authors

The limitaion for the paper can not solved by the revison, and authors seems more like novelty but the scientific.

Author Response

Dear Reviewer,

Thank you for your remarks, but we carefully revised and corrected the paper according to all your comments.

All co-authors are well-known scientists who have published a high number of articles, and this indicates that the authors are not beginners.  Furthermore, our team are deeply interested in microbiota research, has written more than 20 articles on this yield in the Web of Sciences, and also are member and expert in microbiota research and gastroenterology.

Sincerely,

Authors